# Cotinine as a Sentinel of Canine Exposure to Tobacco Smoke

**DOI:** 10.3390/ani13040693

**Published:** 2023-02-16

**Authors:** Debora Groppetti, Giulia Pizzi, Alessandro Pecile, Valerio Bronzo, Silvia Michela Mazzola

**Affiliations:** Department of Veterinary Medicine and Animal Sciences, Università degli Studi di Milano, 26900 Lodi, Italy

**Keywords:** cotinine, dog, hair, serum, smoke, tobacco

## Abstract

**Simple Summary:**

Smoking is a global health problem, recognized as being responsible for increased risk for many diseases. Pets cohabiting with smoking owners may be exposed to tobacco by inhalation, absorption, or ingesting residual smoke particles. Cotinine is a product of the endogenous metabolism of nicotine, and it is used as a biomarker of environmental cigarette smoke exposure in humans. In particular, cotinine in biofluids (blood, saliva, and urine) and hair provides information on short- and long-term smoke exposure, respectively. Despite the considerable evidence of the harmful effects of active and passive tobacco smoke, few studies have explored the relationship between secondhand smoke exposure and cotinine in dogs. This study aimed to measure cotinine concentration in the serum and hair of dogs that were exposed to the owner’s tobacco smoke and to compare it with that of unexposed dogs. Moreover, the influence of exposure intensity (number of cigarettes), age, weight, and sex on cotinine concentration was explored. Cotinine was significantly higher in exposed than unexposed dogs in serum and fur. A sex difference in the concentration of cotinine was also evidenced. These results confirmed the role of cotinine as a sentinel of cigarette smoke exposure in dogs with a different sex sensibility.

**Abstract:**

The adverse health effects of both active and passive tobacco smoke have been well-known in humans for a long time. It is presumable that even pets, which intimately share the owner’s lifestyle, may be exposed to the same risks. This study aimed to detect and quantify cotinine (a metabolite of nicotine) in the serum and hair of dogs using a specific commercial ELISA immunoassay kit. A total of 32 dogs, 16 exposed and 16 unexposed to the owner’s smoke, were enrolled. The cotinine concentration was higher in the exposed than the unexposed group in both matrices (*p* < 0.001), with greater values in serum than in hair (*p* < 0.001). Exposed bitches had higher hair cotinine than male dogs (*p* < 0.001). Conversely, serum and fur cotinine concentrations were lower in female than male dogs of the unexposed group (*p* < 0.01). The exposure intensity, age, and weight of the dogs did not affect cotinine concentrations. A cut-off value of 2.78 ng/mL and 1.13 ng/mL cotinine concentration in serum and fur, respectively, was estimated to distinguish between the exposed and unexposed dogs. Cotinine was confirmed as a valuable marker of passive smoking also in dogs. Although owners do not perceive secondhand smoke as a risk for their dogs, greater awareness should be advisable, especially in pregnant animals.

## 1. Introduction

Active and passive tobacco smoke is a significant cause of preventable disability and early death in humans [1,2,3]. There is no safe exposure to tobacco smoke, which can cause various types of cancer, besides deleterious effects on the nervous, endocrine, immune, respiratory, and cardiovascular systems [1,4]. Secondhand smoke (SHS), also called environmental tobacco smoke, is the combination of side stream smoke from the burning end of a cigarette and the mainstream smoke that is exhaled by smokers [1]. Thirdhand smoke (THS) refers to contaminant particles persisting after secondhand tobacco smoke has been emitted into the air and becomes embedded in materials after smoking a cigarette [5]. Thus, it refers to chemicals that adhere to surfaces from which can then be released back into the air, undergo chemical transformations, or accumulate.

Cotinine, the tobacco alkaloid nicotine’s primary metabolite, acts as a weak agonist of nicotinic acetylcholine receptors (nAChRs). Evidence indicates that cotinine produces diverse neuropharmacological and behavioral effects, and may contribute to nicotine-induced tobacco use, abuse, and dependence [4]. In humans, cotinine is considered a biological biomarker for tobacco smoke exposure, whether active or passive [3,6,7]. Indeed, an average of 70–80% of absorbed nicotine is converted to cotinine [4]. Blood, urine, or saliva specimens provide a short lookback window of the past few days’ exposures, while hair is a long-term matrix that allows the detection of cotinine accumulating up to approximately three months or more as each cm of scalp hair reflects about one month of tobacco exposure [8]. Cotinine is a specific marker of nicotine intake that is not influenced by other pollutants [7].

Pets and humans are increasingly becoming symbiotics by sharing spaces, habits, and food, thus being exposed to the same environmental risks and diseases. Over the past two decades, the harmful effects of secondhand tobacco smoke exposure on children and adults has been widely discussed and underlined through public health campaigns but no emphasis has been placed on the risks that household pets may encounter [9]. So far, dogs have been used as a model for investigating the carcinogenicity of tobacco smoke exposure in humans [10,11]. However, no data on secondhand tobacco smoke-related cotinine concentrations in the serum and hair of dogs exist. This study aimed to explore if household dogs that were exposed to tobacco smoke express cotinine in serum and hair. Reference values of cotinine in dogs may allow discriminating which animals are exposed, knowingly or not, and to correlate it with pathological conditions such as tumors, cardiovascular, and reproductive dysfunctions. For this purpose, serum and hair cotinine concentration was compared in two dog groups: exposed to the owners’ tobacco smoke (EX) and unexposed (UE) ones. Furthermore, parameters such as age, weight, sex, and intensity of exposure to tobacco smoke were explored.

## 2. Materials and Methods

This study is part of a project on environmental factors influencing canine reproduction and was approved by the Ethics Committee of the Università degli Studi di Milano (ethical approval code: OPBA_161_2019). All the owners were informed in detail of the objectives and methods of the research and consented to participation by issuing full informed consent. 

### 2.1. Animals

The study was carried out on 32 clinically healthy dogs that were recruited through the Hospital of Veterinary Medicine of the University of Milan. The animals were evaluated according to their medical history, the absence of any previous illness, and the absence of drugs or dietary supplements. General clinical examinations were performed for all dogs with recording details, which are summarized in Table 1.

Dogs were divided into two groups based on the smoking habits of the owners. In group EX, sixteen dogs (six females and ten males) living with smoking owners and thus passively exposed to tobacco smoke were included. The sixteen dogs that were enrolled in group UE (6 females and 10 males) lived with non-smoking owners and were unexposed to secondhand smoke. 

Dog owners were defined as smokers in case of their indoor consumption of at least one cigarette per day in the last two months. The intensity of secondhand smoke exposure was determined semi-quantitatively and scored as follows: “+” one cigarette per day, “++” up to 10 cigarettes per day, and “+++” more than 10 cigarettes per day. Only household dogs were included. Electronic cigarettes were excluded.

### 2.2. Blood and Hair Sample Collection

Blood samples (1.5 mL) were collected from the cephalic vein into serum tubes (Vacutest, Securlab, Roma, Italy), immediately centrifuged at 3500× *g* for 5 min, and stored at −20 °C until laboratory analysis. All sampling was collected during morning hospital activity, between 9 and 12 a.m.

Hair samples were obtained by shaving the forelimb area before collecting the blood sample, close to the skin, with a clipper. Hair was stored in a paper envelope, in a dry place, until further laboratory analysis.

### 2.3. Laboratory Procedure

The hair cotinine extraction process was performed following the procedure that was described by Gunay et al. [12]. Briefly, one hundred milligrams of hair samples were washed twice with double-distilled water, and the sample was allowed to dry overnight. The washed and dried hair was powdered in a ball mill (Retsch MM400, Retsch-Allee 1–5, Haan, Germany), and 40 mg of shredded hair was weighed into vials; then, 2.5 mL of hair extraction buffer (Immunalysis Corporation, Pomona, CA, USA) was placed in each sample and extracted in an ultrasonic water bath at 25 °C for 30 min. After extraction, 200 μL of neutralization buffer (Immunalysis Corporation, Pomona, CA, USA) was placed in each a microcentrifuge tube and then 50 μL of solution was transferred onto an enzyme-linked immunosorbent assay (ELISA).

The hair and serum cotinine samples were analyzed following the manufacturer’s instructions, using a commercially available pan-specific assay kit (competitive inhibition enzyme immunoassay) designed to accurately measure the cotinine levels in various sample matrices (CET058Ge, Cloud Clone, Katy, TX, USA). Serum samples were used as neat (undiluted), according to pre-experiment results and assay manufacturer recommendations. 

Samples were aliquoted into wells in duplicate (50 µL), and absorbance was measured using a wavelength of 450 nm in a microplate reader (Multiskan EX, LabSystem, Thermo Fisher Scientific, Milan, Italy). The concentration was than calculated by regression analysis according to the relevant standard curves (range 50 pg/mL to 617.3 pg/mL). The mean recovery was 98% ± 7.0. The laboratory researcher was blinded to the hypotheses and conditions.

### 2.4. Statistical Analysis

The data were analyzed using a statistical program (IBM SPSS 28.0, Armonk, NY, USA). Descriptive statistics for quantitative and continuous variables were expressed as mean, standard deviation, and mean standard error. For statistical purposes age (≤2 years; >2 years and ≤5 years; >5 years) and body weight (≤20 kg; <30 kg; ≥30 kg) were stratified into three categories each, and both were analyzed as continuous and categorical variables. Cotinine concentration between EX and UE groups was compared using the *t* test for independent samples in serum and the Mann–Whitney U test in hair. A comparison of serum and hair cotinine was performed with a Wilcoxon test for two matched samples. Intensity of exposure to tobacco smoke was correlated to cotinine concentration in serum using the one-way ANOVA test with post hoc Bonferroni comparison and in hair using the Kruskal–Wallis test for independent samples and post hoc pairwise comparison. Age was analyzed with the one-way ANOVA test with post hoc Bonferroni comparison. Body weight was analyzed with the Kruskal–Wallis test for independent samples and post hoc pairwise comparison. Sex influence was evaluated using the Mann–Whitney U test. The use of parametric or non-parametric test was related to the assessment of normality of data distribution using the Shapiro–Wilk test.

## 3. Results

Cotinine was detectable in all the serum and hair samples of both exposed and unexposed dogs. A positive correlation was recorded between serum and hair cotinine concentration of the total caseload (*p* = 0.001; Figure 1), with higher serum values than hair. Cotinine concentration varied from 0.5 to 15.4 ng/mL (4.6 ± 4.01 ng/mL) in serum and 0.5 to 15 ng/mL in hair (2.7 ± 3.3 ng/mL).

Serum cotinine significantly differed between the EX and UE group (*p* < 0.001), with higher serum values in dogs living with smoking owners (7.4 ± 3.7 ng/mL) than in unexposed ones (1.5 ± 0.7 ng/mL; Figure 2).

Similarly, hair cotinine was higher in the exposed (4.4 ± 3.9 ng/mL) than in the unexposed dogs (0.9 ± 0.3 ng/mL; *p* < 0.001). Cut-off values of cotinine concentration to distinguish between exposed and unexposed dogs have been estimated at 2.78 ng/mL in serum and 1.13 ng/mL in hair (Figure 3).

In the EX group, bitches showed cotinine concentrations that were higher (7.3 ± 5.2 ng/mL) than male dogs (2.7 ± 0.9 ng/mL; *p* < 0.001) in hair, while no sex differences were recorded in serum. In the EU group, bitches had lower cotinine concentrations (0.8 ± 0.4 ng/mL in serum; 0.7 ± 0.1 ng/mL in hair) than males (1.9 ± 0.4 ng/mL in serum; 1.1 ± 0.3 ng/mL in fur) in both matrices (*p* < 0.01). The other parameters, such as the intensity of tobacco smoke exposure, age, and weight, did not influence the cotinine concentration in serum and hair (Table 2).

## 4. Discussion

As in children, even for pets, exposure to environmental tobacco smoke may occur through the inhalation of airborne smoke, transdermal absorption, and ingestion of residues that are deposited on their fur during oral grooming [13]. Raising awareness among smoking pet owners about the potential harm passive smoking could inflict on their companion dogs is not a negligible factor, not only in terms of preventing smoking-related diseases, but also in protecting animal well-being.

Cotinine is the major metabolite of nicotine that is produced endogenously from nicotine intake. Cotinine concentrations are not influenced by common pollution exposure; thus, its detection supports the systemic absorption of nicotine [14]. This aspect indicates that cotinine is perfect for understanding the levels of tobacco exposure. Although only a few data exist, the similarity between the dog and the human in cotinine metabolism has been reported [15]. 

To our knowledge, this is the first study that evaluates cotinine concentration in dogs that are exposed to environmental tobacco smoke. To date, only one other study that reports canine plasma cotinine evaluation has been published. The experimental model of the other study necessitated permanent tracheostomas via cuffed tracheostomy tubes connected to a smoking machine to administer cigarette smoke [16]. This experimental model cannot be considered an actual model of smoke exposure in pets. 

Few other canine studies investigated nicotine concentration in hair [17,18], and cotinine concentration in urine [13]. Currently, urine cotinine is one of the most widely used biomarkers for environmental tobacco smoke assessment in humans, despite significant individual variability in its excretion and a relatively short half-life [19]. Similarly, cotinine is rapidly eliminated from serum after exposure to tobacco smoke [20]. Cotinine in saliva has analogous disadvantages other than being concentrated in the salivary glands, which is misleading with increased concentrations [19]. In addition, cotinine concentrations in these matrices are much lower in passive than active smokers, making them less suitable for assessing second-hand exposure [20]. In contrast, hair samples may allow monitoring of cotinine bioaccumulation due to long-term exposure; this matrix is less affected by daily exposure and metabolism variations and is easily collected and stored at room temperature for up to five years [12,20]. To date, it is unclear whether melanin can interfere with cotinine uptake in hair as there are conflicting results, while shampoo washing does not seem to affect it [8,21,22]. The small number of dogs, together with the heterogeneity of fur colors and types in our caseload, prevent us from speculating. 

As expected, in accordance with what has been published in human studies [6,12,22,23,24], the present study found that dogs living with smoking owners had significantly higher serum and fur cotinine concentrations than dogs that were not exposed to cigarette smoke. The mean cotinine value was 7.4 ng/mL in serum and 4.4 ng/mL in dogs’ hair with smoking owners, and a cut-off of 2.78 ng/mL in serum and 1.13 ng/mL in hair was estimated to differentiate between exposed and unexposed dogs. Our results confirmed the usefulness of cotinine measurements in serum and hair as a marker of smoke exposure in dogs. Considering dogs’ relatively short life expectancy, the harmful effects of second-hand smoke in pets can be an alarm bell for humans. Infectious diseases, antimicrobial resistance, allergies, food intolerance, and tumors have been linked to smoke exposure that interferes with the immune system and introduces mutations in tumor suppressor genes [24,25,26,27,28,29]. It would be interesting to explore the role of cotinine in such pathologies also emerging in veterinary medicine.

In humans, cotinine concentration is positively correlated with the number of hours of passive tobacco exposure or the number of cigarettes actively smoked [30,31]. We found no differences in cotinine concentration according to the intensity of tobacco exposure. However, neither the duration of exposure nor active smoking was considered in our study; besides, only one dog was exposed to low-intensity smoke (one cigarette per day). The influence of cigarette brands, nicotine content, and the presence or absence of a filter on cotinine concentration cannot be excluded and require further research.

Differences in nicotine and cotinine pharmacokinetics between children and adults are reported by some authors [3,25], while others observed no effect of age on cotinine concentration among adults [20,32]. In agreement with the latter, age did not influence cotinine concentration in adult dogs, although data on puppies were not collected.

In the literature, the link between smoking and body weight refers only to weight loss and weight control due to its effect on metabolism and appetite [33,34]. To the best of the authors’ knowledge, no data exist on a possible different cotinine uptake in relation to body weight. Although without statistical significance, we recorded lower levels of cotinine in heavy dogs (from 30 kg upwards), especially in the serum but also in the hair, compared to smaller dogs. We speculate that the low respiratory rate of large size dogs [35] could affect cotinine concentrations. Once again, although caution should be exercised when generalizing the current results, no age-related differences in cotinine concentration emerged in dogs. 

Gender medicine is a new approach to healthcare aimed at recognizing and analyzing the differences arising from sex in several aspects, including pharmacokinetics and pharmacodynamics, in response to drug exposure [36,37]. In this context, sex comparison also seems to play an important role in passive smoking. In humans, there are few and mixed findings on sex differences in cotinine concentrations. Mean cotinine levels are higher in men than in women in some studies, whereas women have higher levels in others [38]. Faster hair growth in women than in men and hormonal involvement may account for some sex differences in humans [8] and could also be speculated in dogs. Indeed, in our study, a sex difference was recorded with an opposite trend of cotinine concentrations in exposed and unexposed bitches compared to males, suggesting a greater sensitivity to tobacco smoke exposure in female dogs. Studies in rats have come to similar conclusions, showing higher levels of nicotine in females [39]. This frailty should be carefully considered in relation to the risk to pregnant and lactating bitches that are exposed to second-hand smoke.

Finally, it is noted that some aspects were not considered in this pilot study and deserve further investigation. For example, ethnicity or breed may be involved in cotinine metabolism variability, as may the type and texture of hair [7,8,40]. Despite hair growth in mammals differing from humans due to the unsynchronized growth cycles [41], similar morphological characteristics and compositions to canine bulge area as well as similarities in follicular biomarker expression, make the dog a model close to humans [42]. Albeit modestly, urinary cotinine was reported to be related to fur length in dogs [13]. Furthermore, due to less filtration of cigarette smoke occurring in the nasal cavity of brachycephalic than mesocephalic and dolichocephalic dogs, nose length is regarded as a potential risk factor for exposure to tobacco smoke [13]. Moreover, the quality and quantity of food could affect cotinine uptake [28] as well as dogs’ activity.

## 5. Conclusions

The result of this preliminary study showed the exposure to second-hand smoke of dogs living with smoking owners. Cotinine has been confirmed to be a reliable marker to discriminate between exposed and unexposed dogs with some sex sensitivity differences. While both serum and hair are suitable matrices for cotinine measurement, hair samples are easier to collect and less expensive to store and provide long-time exposure to tobacco smoke [40]. Despite the increased social function of pets, owners seem to have little awareness of the risks faced by pets that are exposed to smoke. Further studies are needed to investigate the relationship between cotinine uptake and some variable such as breed, age, weight, diet, and hair constitution.

## Figures and Tables

**Figure 1 animals-13-00693-f001:**
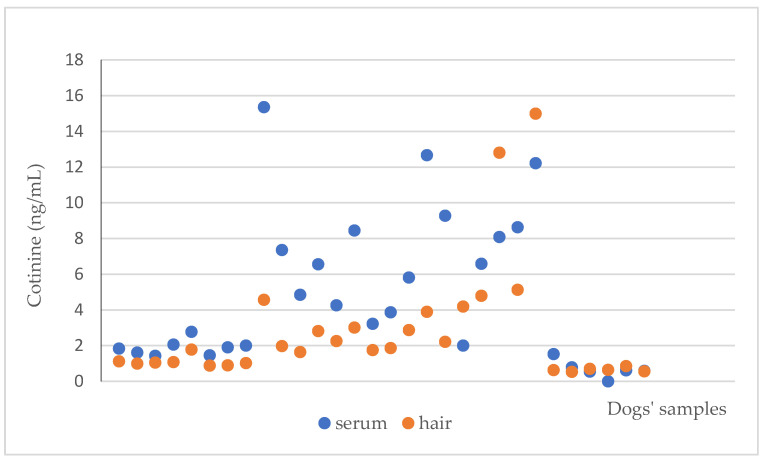
Serum and hair cotinine concentrations in the total caseload.

**Figure 2 animals-13-00693-f002:**
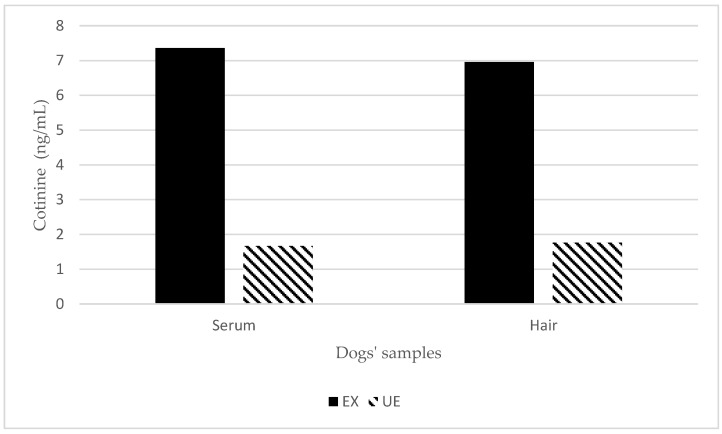
Serum and hair cotinine concentrations in exposed and unexposed dogs. EX means dogs that were exposed to the owner’s tobacco smoke; UE means dogs that were unexposed to the owner’s tobacco smoke.

**Figure 3 animals-13-00693-f003:**
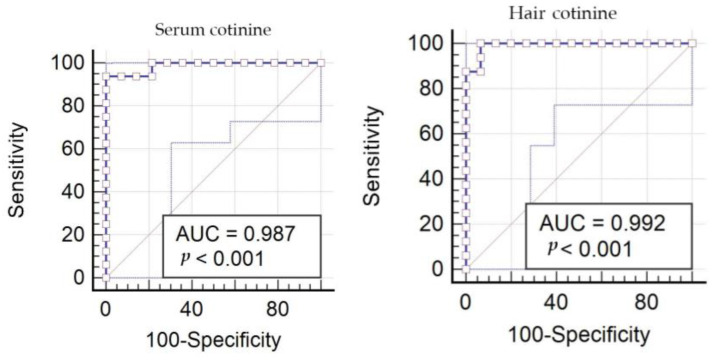
Cutoff values of cotinine concentrations in serum and hair.

**Table 1 animals-13-00693-t001:** Main data of dogs that were enrolled in this study.

	Breed	Age (Years)	BW (kg)	Sex	Smoke Exposure	Exposure Intensity	Group
1	Entlebucher Mountain Dog	2.5	27.4	Male	Yes	+++	EX
2	Kurzhaar	5	39.7	Male	Yes	++	EX
3	Labrador retriever	2	42.5	Male	Yes	+++	EX
4	Labrador retriever	2	26.5	Male	Yes	+++	EX
5	Labrador retriever	4	37.1	Male	Yes	+++	EX
6	Bassethound	2.5	21.8	Male	Yes	+++	EX
7	Great dane	5.5	77.7	Male	Yes	+	EX
8	Afgan hound	4.5	30.9	Male	Yes	++	EX
9	Pharaoh hound	4	29.2	Male	Yes	++	EX
10	Bassethound	4	28	Male	Yes	++	EX
11	Bassethound	6	23	Female	Yes	+++	EX
12	Bassethound	3.5	24	Female	Yes	+++	EX
13	Bassethound	1.5	21	Female	Yes	+++	EX
14	Bassethound	2	28	Female	Yes	++	EX
15	Mongrel	12	15	Female	Yes	+++	EX
16	Mongrel	11	30	Female	Yes	+++	EX
17	Kurzhaar	2.5	30	Male	No	none	UE
18	Rhodesian ridgeback	7	40	Male	No	none	UE
19	German shepherd	6.5	34.5	Male	No	none	UE
20	Kurzhaar	5	27.5	Male	No	none	UE
21	Bracco italiano	7.5	31.6	Male	No	none	UE
22	German shepherd	4	32.5	Male	No	none	UE
23	Bernese Mountain Dog	3	51.8	Male	No	none	UE
24	Kurzhaar	6.5	28.4	Male	No	none	UE
25	Hovawart	3	31	Male	No	none	UE
26	Mongrel	1.5	16	Male	No	none	UE
27	Belgian shepherd	8	20.7	Female	No	none	UE
28	Bernese Mountain Dog	2	47.5	Female	No	none	UE
29	Mongrel	0.5	10	Female	No	none	UE
30	Bouledogue	2	17.6	Female	No	none	UE
31	Mongrel	0.5	11.6	Female	No	none	UE
32	Mongrel	0.5	10	Female	No	none	UE

BW body weight; “+” means one cigarette per day; “++” means up to 10 cigarettes per day; “+++” means more than 10 cigarettes per day; EX = dogs exposed to the owner’s tobacco smoke; UE = dogs unexposed to the owner’s tobacco smoke.

**Table 2 animals-13-00693-t002:** Cotinine concentrations.

Parameter	Exposed Dogs (EX)		Unexposed Dogs (UE)	
	Cotinine (ng/mL)		Cotinine (ng/mL)	
	serum	hair	serum	hair
	7.4 ± 3.7	4.4 ± 3.9	1.5 ± 0.7	0.9 ± 0.3
Intensity +	4.3 ± 1.3	2.2 ± 0.6	-	-
Intensity ++	9.4 ± 2.9	6.2 ± 5.8	-	-
Intensity +++	7.8 ± 3.9	4.6 ± 3.9	-	-
Age 1	6.5 ± 1.3	5.5 ± 5.03	0.9 ± 0.6	0.7 ± 0.2
Age 2	7.5 ± 4.6	3.1 ± 1.04	2.1 ± 0.4	1.2 ± 0.3
Age 3	8.3 ± 3.5	6.02 ± 6.2	1.6 ± 0.3	0.9 ± 0.2
BW 1	8.6	5.1	0.9 ± 0.6	0.8 ± 0.2
BW 2	8.3 ± 3.9	4.6 ± 3.2	1.7 ± 0.3	0.9 ± 0.2
BW 3	5.9 ± 3.4	4.1 ± 5.3	1.8 ± 0.6	1.1 ± 0.4
Male	7.2 ± 3.9	2.7 ± 0.9	1.9 ± 0.4	1.1 ± 0.3
Female	7.8 ± 3.4	7.3 ± 5.2	0.8 ± 0.4	0.6 ± 0.1

BW body weight; “+” means one cigarette per day; “++” means up to 10 cigarettes per day; “+++” means more than 10 cigarettes per day; EX means dogs that are exposed to the owner’s tobacco smoke; UE means dogs that are unexposed to the owner’s tobacco smoke; Age 1 means ≤2 years; Age 2 means >2 years and ≤5 years; Age 3 means >5 years); BW 1 means ≤20 kg; BW 2 < 30 kg; BW 3 ≥ 30 k.

## Data Availability

All the data that support the findings of this study are available from the corresponding author.

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
