# Peer review of "Cotinine as a Sentinel of Canine Exposure to Tobacco Smoke"

_animals, 2023, doi:10.3390/ani13040693_

Round 1

Reviewer 1 Report

The study aims to measure cotinine concentration in the serum and hair of dogs who owner smokes tobacco products using a more easy way than previous described. This is important because it makes more accessible to test pets living in environments exposed to smoke and could increase the association with other conditions, as well as allowing owners to know the risk to which their pets are exposed.

The strengths of this manuscript are certainly the clear exposition of the project, the details of the methods used, and the logical and fluid discussion of the results obtained.

Minor revisions suggested are:

1)             To potentiate the importance of the study, please specify that Cotinine is the main metabolite of nicotine, in this way it will be underlined that the approach described in the manuscript is the way to go if the goal is to understand the levels of tobacco's exposure.

2)             Since different types of owners are included (+, ++ and +++), it could be appropriate to report the timing of sampling. Indeed, the routine of the smokers could affect the quantification not just in term of numbers of cigarette, but gap-time since the last one. For example, if the blood was collected early morning it is possible that during the night that dog was not full exposed to the same amount of nicotine compared to an other sample where the blood was collected during lunch time.

3)             Figure 1 doesn’t report any value or indication in the X axis.

4)             In Figure 2 “Serum” and “Hair” cover portion of the bar graph, please format according.

5)             There is a classification for BW, but all BW are called in the same way. They should be differentiated in the lettering (e.g., BW I, BW II, BW III).

6)             In Table 2 is reported an interesting increase of ng/ml of cotinine across age1-2-3, but the data are not fully discussed in the discussion. Is it possible to speculate that is correlated with the accumulation in the body in relation of breathing rate?

7)             The authors in the discussion correlated their study with human studies, specifically for the sex difference, there are in literature information from the basic science portion, using small animal as rodents, that can help the authors to support the interesting difference they observed.

Author Response

Comments and Suggestions for Authors

REV 1

The study aims to measure cotinine concentration in the serum and hair of dogs who owner smokes tobacco products using a more easy way than previous described. This is important because it makes more accessible to test pets living in environments exposed to smoke and could increase the association with other conditions, as well as allowing owners to know the risk to which their pets are exposed.

The strengths of this manuscript are certainly the clear exposition of the project, the details of the methods used, and the logical and fluid discussion of the results obtained.

We sincerely appreciate all of your valuable comments and suggestions, which helped us improve the manuscript's quality. Hint insights provided by the reviewer were received and developed in the discussion. Thank you again for your help in improving our work.

Minor revisions suggested are:

1)             To potentiate the importance of the study, please specify that Cotinine is the main metabolite of nicotine, in this way it will be underlined that the approach described in the manuscript is the way to go if the goal is to understand the levels of tobacco's exposure.

This aspect has been underlined in the text at LL 230-234.

2)             Since different types of owners are included (+, ++ and +++), it could be appropriate to report the timing of sampling. Indeed, the routine of the smokers could affect the quantification not just in term of numbers of cigarette, but gap-time since the last one. For example, if the blood was collected early morning it is possible that during the night that dog was not full exposed to the same amount of nicotine compared to another sample where the blood was collected during lunch time.

We agree. All sampling was collected between 9 and 12 AM, and we have added this information at Line 118. Most owners reported smoking mainly in the evening after returning from work, but we did not collect this data for the study. Thanks for your suggestion; it's an important consideration for future research.

3)             Figure 1 doesn’t report any value or indication in the X axis.

We thank the Referee for reporting this lapse. Figure 1 has been edited.

4)             In Figure 2 “Serum” and “Hair” cover portion of the bar graph, please format according.

We thank the Referee for reporting this lapse. Figure 2 has been edited.

5)             There is a classification for BW, but all BW are called in the same way. They should be differentiated in the lettering (e.g., BW I, BW II, BW III).

Thank you for bringing to our attention the oversight in our manuscript. We have now included the information you requested in Table 2.

6)             In Table 2 is reported an interesting increase of ng/ml of cotinine across age1-2-3, but the data are not fully discussed in the discussion. Is it possible to speculate that is correlated with the accumulation in the body in relation of breathing rate?

That is a very interesting hypothesis. We discussed this at LL “282-285.

7)             The authors in the discussion correlated their study with human studies, specifically for the sex difference, there are in literature information from the basic science portion, using small animal as rodents, that can help the authors to support the interesting difference they observed.

We thank the reviewer for this comment. We have added in discussion at LL 298-290 studies in rats reporting similar results.

Reviewer 2 Report

This paper is potentially interesting but there are some issues that should be carefully addressed by authors before making the paper suitable for publication in the Animals.

Line 31: Please add the cut-off value.

Line 42: Please replace STS with SHS.

Lines 56-57: Hair allows detection time depending of hair length, every cm corresponds to one month exposure. So, it allows detection more than three months, if the hair is longer than 3 cm.

Lines 63-65: Reference is missing.

Line 72: I suggest to delete word ‘’clinical’’.

Line 104: Did you use whole fur length or cut the same length for every sample. Please describe.

Section 2.3. Did manufacturer report any cross-reactivity? If so, please describe.

Line 149: Why did you present results in hair in ng/mL (mL of what?). I suggest to recalculate in ng/mg of hair. There is no point to compare the results in serum with hair.

Section 4: You should briefly discuss the difference in metabolism between human and dog as well as in hair growth.

Author Response

Comments and Suggestions for Authors

REV 2

This paper is potentially interesting but there are some issues that should be carefully addressed by authors before making the paper suitable for publication in the Animals.

The authors thank the Reviewer for the positive comments and the suggestions that contribute to improving the manuscript.

Line 31: Please add the cut-off value.

Done

Line 42: Please replace STS with SHS.

Thanks for reporting this lapse.

Lines 56-57: Hair allows detection time depending of hair length, every cm corresponds to one month exposure. So, it allows detection more than three months, if the hair is longer than 3 cm.

Thanks for this clarification which was added in the text.

Lines 63-65: Reference is missing.

The reviewer is right. References have been added.

Line 72: I suggest to delete word ‘’clinical’’.

We agree, the word “clinical” has been deleted.

Line 104: Did you use whole fur length or cut the same length for every sample. Please describe.

We use whole fur length, and did not measure the length of individual hair samples. Instead, we collected hair from the front limb area where we later performed the blood draw. In all dogs included in the study, the hair in that area was short, and therefore, we deemed it unnecessary to measure its length.

Section 2.3. Did manufacturer report any cross-reactivity? If so, please describe.

The manufacturer declared that the Elisa assay we used  has high sensitivity and excellent specificity for detection of cotinine. No significant cross-reactivity or interference between cotinine and analogues was observed. The manufacturer also declared that limited by current skills and knowledge, it is impossible for them to complete the cross- reactivity detection between cotinine and all the analogues, therefore, cross reaction may still exist.

Line 149: Why did you present results in hair in ng/mL (mL of what?). I suggest to recalculate in ng/mg of hair. There is no point to compare the results in serum with hair.

Thank you for the point. We used the methodology described in the literature for the quantification of cotinine in hair, which involves the extraction of hair cotinine into a liquid substrate. We could have indeed converted the values obtained through the ELISA kit (expressed in ng/ml) into ng/mg of hair. However, we decided not to convert the values to better present them graphically.

Section 4: You should briefly discuss the difference in metabolism between human and dog as well as in hair growth.

Very few data exist in the literature. We have reported these studies and discussed them in the text.

Reviewer 3 Report

The paper “Cotinine as a sentinel of canine exposure to tobacco smoke” by Groppetti et al. describes cotinine levels in hair of dogs living with smoking and non-smoking humans. Cotinine levels are regarded as a good marker of smoke exposure both in humans and laboratory animals, and sometimes even the only possibility to determine the real exposure or passive smoking. The paper summarizes experimental work, however, the main aim of the paper, the leading idea of the entire project is not clearly shown.  

I would like to ask the authors why they decided to compare cotinine levels in tobacco smoke-exposed and unexposed dog? What was the purpose of the entire study or project? Are these experiments a part of a larger study dealing with tabaco smoke exposure on a health issue of dogs or any other animal? Could the authors clarify this problem? Dogs are used in preclinical research, the relation between the nicotine metabolite level and exposure is something well known (see citations)?  

Could the authors comment why levels of cotinine were sex-dependent? The present explanation of the problem is not clear.  

Could the authors comment on other possible factors influencing cotinine levels, like maybe the dogs’ activity (Great Dane’s and a Bouledogue’s (outdoor) activity surely varies) or others? 

Minor comments: 

  • Do not use lines in Fig 1 – it represents individual cases and there should not be any connection between the samples; 

  • Are the data means with SEM or SE? 

  • Add SEM/SE to Fig 2; 

  • Please provide figures in better quality; 

  • Use dots and not commas in numbers; 

  • Did you mean in line 168 “UE”? 

Altogether, an extensive discussion of the obtained results together with a more clear description of the aims of the study would enrich this paper.

Author Response

REV 3

We would like to thank the reviewer for the time they have dedicated to our manuscript, we have taken the utmost care with their observations, and have revised the manuscript according to their suggestion.

The paper “Cotinine as a sentinel of canine exposure to tobacco smoke” by Groppetti et al. describes cotinine levels in hair of dogs living with smoking and non-smoking humans. Cotinine levels are regarded as a good marker of smoke exposure both in humans and laboratory animals, and sometimes even the only possibility to determine the real exposure or passive smoking. The paper summarizes experimental work, however, the main aim of the paper, the leading idea of the entire project is not clearly shown. 

To date, cotinine has never been evaluated in dogs, unlike nicotine. In humans and laboratory animals, cotinine has shown to be a highly reliable biomarker of exposure to smoke, including secondhand smoke, far superior to nicotine. To further strengthen its usefulness as a biomarker of smoke exposure, it has not yet been shown that other environmental pollutants (other than nicotine) can influence endogenous cotinine levels. The importance of the study is therefore linked to these factors.

I would like to ask the authors why they decided to compare cotinine levels in tobacco smoke-exposed and unexposed dog? What was the purpose of the entire study or project? Are these experiments a part of a larger study dealing with tabaco smoke exposure on a health issue of dogs or any other animal?

As stated in LL 77-78, this study is part of a project that explored environmental factors that impact canine reproduction, in which various factors were investigated, including cotinine. The first step of the research was to compare exposed and non-exposed dogs to secondhand smoke, in order to highlight if measuring cotinine in serum and hair was a reliable parameter. This is an important piece of information because it could allow owners to know the risk their pet animals are exposed to. Furthermore, it makes testing on pets living in environments exposed to smoke more accessible, making it easier to associate with other pathological conditions.

Could the authors clarify this problem? Dogs are used in preclinical research, the relation between the nicotine metabolite level and exposure is something well known (see citations)?

Currently, there are no studies in literature on the concentration of cotinine in dogs spontaneously exposed to environmental tobacco smoke (see LL 234-238). To date, literature only refers to the measurement of nicotine in dogs. However, the measurement of cotinine is preferable to the measurement of nicotine because cotinine persists in the body for a longer time, characterized by a plasma half-life of about 16 hours (Benowitz and Jacob, 1994).

Could the authors comment why levels of cotinine were sex-dependent? The present explanation of the problem is not clear. 

The referee highlighted an aspect that is still debated in human medicine. We hypothesized that, as occurs in humans, the difference in hair growth and hormonal profile between males and females may be the basis for the differences observed in cotinine concentration in hair (LL 297-298).

Could the authors comment on other possible factors influencing cotinine levels, like maybe the dogs’ activity (Great Dane’s and a Bouledogue’s (outdoor) activity surely varies) or others?

Thank you for pointing out this aspect that we have added in discussion.

Minor comments:

Do not use lines in Fig 1 – it represents individual cases and there should not be any connection between the samples;

We agree with the reviewer. The figure has been modified accordingly.

Are the data means with SEM or SE?

Data were expressed as mean, standard deviation and mean standard error (LL 147).

Add SEM/SE to Fig 2;

Mean and standard deviation are provided just before Figure 2 (LL 175-176). If the reviewer deems it important, we can add SEM/SE in Figure 2.

Please provide figures in better quality;

Done

Use dots and not commas in numbers;

Figure 3 is generated by the software, which uses commas instead of numbers. if the referee prefers, we can modify the figure with Photoshop.

Did you mean in line 168 “UE”?

In line 168, we refer to the total caseload, including both EX and UE. We have clarified this concept in the text.

Altogether, an extensive discussion of the obtained results together with a more clear description of the aims of the study would enrich this paper.

The discussion was implemented, following the referee suggestion.

Round 2

Reviewer 2 Report

The manuscript is improved.

Reviewer 3 Report

Dear Authors,

Thank you very much for the explanations in the text. I believe, there is nothing more to add or to correct. Just have a look on small mistakes like k. instead of “kg” in line 227 but these are just details. Nevertheless, if the software gives decimal values with commas, maybe it would be an idea to change its settings as in the English language, dots refer to decimal values whereas commas to thousands. But it’s a common problem of researchers from some non English-speaking countries, and it would be nice to change it even in a graphical software. Changes between dots and commas can be misleading in case of i.e. ELISA studies when somebody is not reading carefully into the text. But it’s up to you.